



# Downscaling of air pollutants in Europe using uEMEP_v6

Qing Mu[1], Bruce Rolstad Denby[1], Eivind Grøtting Wærsted[1], and Hilde Fagerli[1]

[1]The Norwegian Meteorological Institute, Henrik Mohns Plass 1, 0313, Oslo, Norway

**Correspondence:** Qing Mu (qingm@met.no)

**Abstract.** The air quality downscaling model uEMEP and its combination with the EMEP MSC-W chemical transport model are used here to achieve high-resolution air quality modeling at street level in Europe. By using publicly available proxy data, this uEMEP/EMEP modelling system is applied to calculate annual mean $NO_2$, $PM_{2.5}$, $PM_{10}$ and $O_3$ concentrations for all of Europe down to 100 m resolution and is validated against all available Airbase monitoring stations in Europe at 25 m resolution. Downscaling is carried out on annual mean concentrations, requiring special attention to non-linear processes, such as $NO_2$ chemistry, where frequency distributions are applied to better represent the non-linear $NO_2$ chemistry. The downscaling shows significant improvement in $NO_2$ concentrations where spatial correlation has been doubled for most countries and bias reduced from -46% to -18% for all stations in Europe. The downscaling of $PM_{2.5}$ and $PM_{10}$ does not show improvement in spatial correlation but does reduce the overall bias in the European calculations from -21% to -11% and from -39% to -30% for $PM_{2.5}$ and $PM_{10}$ respectively. There is improved spatial correlation in most countries after downscaling of $O_3$, and a reduced positive bias of $O_3$ concentrations from +16% to +11%. Sensitivity tests in Norway show that improvements in the emission and emission proxy data used for the downscaling can significantly improve both the $NO_2$ and PM results. The downscaling development opens the way for improved exposure estimates, improved assessment of emissions as well as detailed calculations of source contributions to exceedances in a consistent way for all of Europe at high resolution.

## 1 Introduction

The EMEP Meteorological Synthesizing Centre - West (EMEP MSC-W) at the Norwegian Meteorological Institute has been developing and implementing a downscaling methodology to enhance the capabilities of the EMEP MSC-W chemical transport model (Simpson et al., 2012, 2020) (hereafter the EMEP model). This downscaling model is known as uEMEP (urban EMEP) and can achieve high-resolution air quality modelling down to 100 m for entire countries (Denby et al., 2020). Even though the methodology is referred to as 'downscaling', uEMEP is actually an independent Gaussian plume modelling system which is added as post-processing to the EMEP model. This makes the modelling similar to other local-scale air quality models and allows for a good physical representation of air quality concentrations.

uEMEP was first reported in the 2016 EMEP status report (Denby and Wind, 2016). Since then uEMEP has been further developed and operationally implemented in the Norwegian Air Quality Forecasting System (Miljodirektoratet, 2020b) as well as providing air quality data, maps and information to Norwegian municipalities through the Air Quality Expert Service (Miljodirektoratet, 2020a). These model applications and validations are described in detail in Denby et al. (2020).





The long-term aim of the uEMEP/EMEP modelling system is to extend uEMEP to cover all the EMEP model domain, so that we can have air quality modelling at street-level all over Europe. Previous air quality modeling across Europe could not reach such a high resolution of 100 m (Sofiev et al., 2015; Menut et al., 2013), or the street-level air quality modeling studies were limited to individual cities (Stocker et al., 2012; Kim et al., 2018). The uEMEP/EMEP modelling system is now established in Norway, where access to good quality emission related data is available. Unfortunately, the same quality of high-resolution emission data that is available in Norway is not directly available for all of Europe. Many countries have suitable high-resolution data but these are not readily accessible for use. In order to implement uEMEP for all of Europe then proxy data that can be used to redistribute emissions to fine scales are required. Three datasets are available for all of Europe, also globally, and have been used to enable the high-resolution modelling in Europe: OpenStreetMap (OSM) (OpenStreetMap contributors, 2020) for redistributing road traffic emissions, population data from Global Human Settlement (GHS) (Schiavina et al., 2019) gridded to 0.0025 degrees for redistributing residential heating emissions, and Automatic Identification System (AIS) (Kystverket, 2020) data for shipping emissions gridded to 0.0025 degrees. These datasets allow downscaling of the traffic, residential heating and shipping emission sources. All other sources are not included in the downscaling.

Results of the European modelling for $NO_2$, $PM_{2.5}$, $PM_{10}$ and $O_3$ are presented as example maps in Sect. 3, validation against Airbase stations is in Sect. 4 and results of a number of sensitivity studies are reported in Sect. 5.

## 2 Methodology

Downscaling with uEMEP applies the following methodology:

- Calculations are made using the EMEP model for all of Europe in a similar way to the official EMEP model calculations but with the additional output of the EMEP local fractions (EMEP Status Report 1/2017, 2017; Wind et al., 2020)

- uEMEP is implemented as a post-processing routine to the annual mean output from the EMEP model. EMEP emission grids per sector and per compound are redistributed onto high-resolution sub-grids using the emission proxies

- uEMEP then calculates the local dispersion from these sub-grid emissions using a dispersion kernel within a region defined by 2 x 2 EMEP grids

- uEMEP removes the local fraction contribution from the EMEP grid results and replaces these with the uEMEP sub-grid results

- Resolution of the sub-grids varies according to the application but maps are made at 100 m and calculations at monitoring sites are made at 25 m.

### 2.1 EMEP model implementation

The EMEP model setup follows that in EMEP/MSC-W et al. (2020). Model version rv4.35 is used in this study, with a horizontal resolution of 0.1°x 0.1°and 20 vertical layers (the lowest layer height of approximately 50 m). The model domain





covers the geographic area between 30°N-82°N latitude and 30°W-90°E longitude. The simulation year is 2018. The EMEP model calculates and outputs 'local fraction' used as input of the uEMEP downscaling in order to remove double counting of emissions (Denby et al., 2020; Wind et al., 2020).

The meteorological data is taken from the Integrated Forecast System (IFS) of the European Centre for Medium-Range Weather Forecasts (ECMWF), with the version IFS Cycle 40r1 (ECMWF-IFS cy40r1). The emission inventory for 2018 is based on the official data submissions to EMEP Centre on Emission Inventories and Projections (Pinterits et al., 2020) in 2020, in which the PM emissions from the residential combustion sector (GNFR C) are replaced by a bottom-up estimate of TNO (Denier van der Gon et al., 2020; Fagerli et al., 2020) for 2017. This TNO dataset should represent an improved estimate of

residential combustion emissions of PM, accounting for condensable organics in a consistent way.

## 2.2   uEMEP model implementation

The uEMEP model is described in a recent publication (Denby et al., 2020). In that article the Norwegian forecast application of uEMEP is described where hourly downscaling using bottom-up emission inventories is carried out. For the European application calculations are made on annual mean data, creating air quality maps for Europe down to 100 m resolution and

calculating concentrations at Airbase stations positions down to 25 m.

Downscaling is carried out in the following way. EMEP grid emissions per sector and per source are redistributed to uEMEP sub-grids using the proxy emission data described in Sect. 2.3. These emission sub-grids are dispersed using a rotationally symmetric Gaussian dispersion kernel (Denby et al., 2020), given an initial plume size and height. These parameters are provided in Table 1. The initial horizontal plume size is determined by the size of the sub-grid. The Gaussian dispersion

parameters used are based on the $K_z$ dispersion methodology described in Denby et al. (2020) but adapted to the rotationally symmetric dispersion kernel. The local fraction contribution from the EMEP model is removed and replaced with the sub-grid dispersion calculation from uEMEP, thus avoiding double counting of emissions.

**Table 1.** Initial dispersion ($\sigma_{z0}$) and emission height ($h_{emis}$) for the three downscaled sources

| Source | Initial dispersion ($\sigma_{z0}$) | emission height ($h_{emis}$) |
|---|---|---|
| Traffic (GNFR6) | 2 m | 1 m |
| Residential heating (GNFR3) | 10 m | 15 m |
| Shipping (GNFR7) | 15 m | 70 m |

Downscaling with uEMEP occurs only for primary emissions within a specified 'local' area surrounding each uEMEP sub-grid. This is referred to as the uEMEP 'moving window'. For these simulations this area corresponds to two EMEP grids,

i.e. within an area that is $\pm\ 0.1^o$ in both latitude and longitude. $NO_2$ is calculated from $NO_X$ and $O_3$ using the same travel time parameterisation described in Denby et al. (2020) but applied to annual mean wind speeds, photo-dissociation rates and concentrations. To account for non-linearity in the $NO_2$ chemistry, when calculating with annual means, an additional





frequency distribution correction factor is implemented, see Sect. 2.4. Annual mean downscaled $O_3$ is also determined using this same parameterisation.

To improve efficiency of the calculations, Europe is split into a number of tiles that cover the European land domain. For the 100 m resolution mapping calculations there are 1097 tiles, each of which is 100 km x 100 km. These tiling regions are shown in Fig. 1.

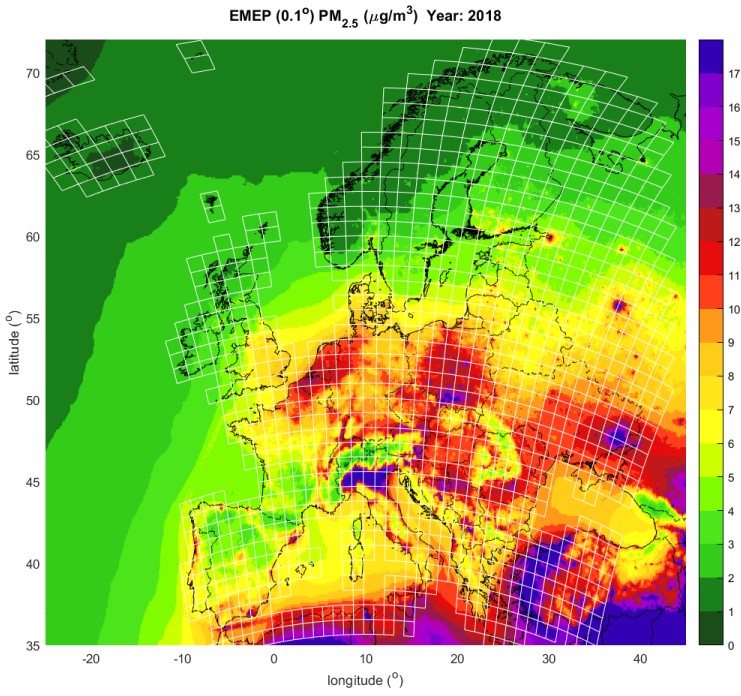

**Figure 1.** Annual mean $PM_{2.5}$ concentrations calculated with the EMEP model at $0.1^o$ for 2018. Shown are the uEMEP tiling regions used in the calculations. In total 1097 100 km x100 km tiles with 100 m resolution are used to model Europe.

## 2.3    uEMEP proxy data

We use road data from OSM to redistribute the traffic emissions. Though the spatial coverage of OSM is very good, it does not
contain actual traffic data. Redistribution of the emission data is achieved by weighting the different road categories provided in OSM. The following road categories are considered: motorway, trunk, primary, secondary, tertiary, unclassified and residential. Each is weighted relative to the other so that emissions can be redistributed and attributed to the road links. Estimates of the weights are based on the representative average daily traffic (ADT) for different road categories for Norwegian average road situations. The weighting currently employed for the redistribution is shown in Fig. 2. It is also worth noting that for major
roads, such as motorways, OSM often represents these as dual carriageways, i.e. as two separate road links. In these cases the weighting of a motorway will be twice that indicated here. Sensitivity tests with alternative weighting, see Sect. 5.2, show the





choice of weighting does impact on the results but that the current choice provides close to optimal spatial correlation when compared to measurements.

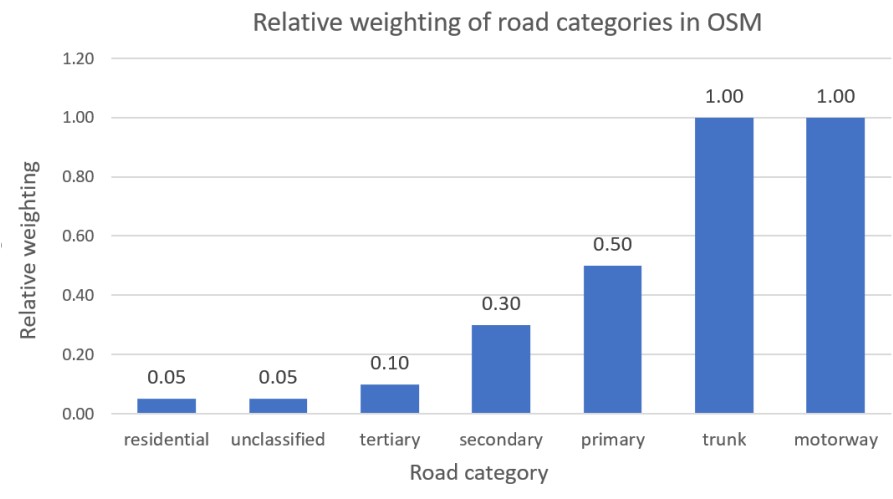

**Figure 2.** Weighting of the OpenStreetMap road categories used to redistribute EMEP emissions for downscaling.

A global population dataset from the GHS is used as the proxy for redistributing residential heating emissions. We choose the
highest available resolution of 9 arcsec ($0.0025^o$) from the year 2015. The coordinate system is WGS84. This dataset indicates the distribution of population as the number of people per cell. A number of alternative formulations of the population proxy, as well as an alternative proxy based on building density, are assessed in Sect. 5.3

AIS data for shipping emissions are provided by the Norwegian Coastal Administration. The raw data, which contains a list of instantaneous point emissions, is averaged over the year 2017 and gridded to $0.0025^o$. Though these data are actual
emissions we still use them as proxy data to redistribute EMEP gridded emissions, to be consistent in the methodology.

### 2.4 uEMEP chemistry parameterisation for annual mean $NO_2$

uEMEP downscales only primary pollutants. It is thus necessary to apply chemistry parameterisations to the $NO_X$ and $O_3$ concentrations to derive $NO_2$. Two methods for doing this are described in Denby et al. (2020). One for hourly concentrations using a weighted travel time parcel method and the other for annual means, which is based on a simple empirical relation-
ship between observed $NO_2$ and $NO_X$. It is desirable to apply the model based chemistry scheme rather than an empirical scheme, however due to the non-linearity of the $NO_2$ chemistry the chemical scheme cannot be directly applied to annual mean concentrations.




To solve this, chemistry is not calculated on just a single annual mean value for $NO_X$ and $O_3$ but on a frequency distribution for these parameters that represent the variability over a year. This can be illustrated for the photostationary case where the

$NO_2$ concentrations can be derived from $NO_X$ and $O_X$ using

$$[NO_2] = \frac{1}{2}\left(([NO_X]+[O_X]+J/k_1) - \sqrt{([NO_X]+[O_X]+J/k_1)^2 - 4[NO_X][O_X]}\right) \tag{1}$$

where the concentrations are annual mean values in molecules/cm$^3$, $k_1$ is the production reaction rate and $J$ the photo-dissociation rate for $NO_2$. If the frequency distribution for the three annual mean variables $[NO_X]$, $[O_X]$ and $J/k_1$ is known

then we can integrate over Eq. (1) using these distributions as weighting functions. An appropriate probability distribution function for the concentrations is the log-normal distribution, which can be written as

$$PDF_x = \frac{1}{x\sigma\sqrt{2\pi}}exp\left(-\frac{(log(x)-\mu)^2}{2\sigma^2}\right) \tag{2}$$

where the log-normal parameters $\mu$ and $\sigma$ are determined from the mean $(m)$ and the standard deviation $(s)$ by

$$\mu = log\left(\frac{m^2}{\sqrt{m^2+s^2}}\right) \text{ and } \sigma^2 = log\left(1+\frac{s^2}{m^2}\right)$$

The frequency distribution of $J$ is not log-normally distributed, since it is dependent on the solar zenith angle $(ZA)$ and various other meteorological parameters, such as cloud cover and water vapour content. The EMEP model uses lookup tables

based on precalculated $J$ values from the Phodis model (Jonson et al., 2000). In order to implement the frequency distribution of $J$ in uEMEP a power law fit is made to the tabulated values. This can be written as:

$$J = C_j cos(ZA)^{-p_j} \tag{3}$$

where $C_j$ is a constant that is normalised out when producing the normalised frequency distribution and $p_j = 0.28$. The standard

deviation of $k_1$, dependent on air temperature, is significantly smaller than for $J$ so it is treated as a constant.

A new value $[NO_2]_{pdf}$ can then be determined using these frequency distributions

$$[NO_2]_{pdf} = \int\!\!\!\int\!\!\!\int\limits_0^\infty [NO_2]\,PDF_{ox}\,PDF_{nox}\,PDF_j\,d[NO_X]\,d[O_X]\,dJ \tag{4}$$





and a correction term showing its difference from the mean is defined as

$$f_{no2,pdf} = \frac{[NO_2]_{pdf}}{[NO_2]_{mean}} - 1 \tag{5}$$

When calculating in three orthogonal dimensions then it is assumed there is no correlation between the variables.

To implement this procedure the standard deviation $s$ must be known for $NO_X$ and $O_X$. Values for $s_{nox}$ and $s_{ox}$ have been derived from earlier model calculations for Norwegian stations. Linear regression provides robust values for $s_{ox}/m_{ox}$ and $s_{nox}/m_{nox}$ of 0.21 and 1.14 respectively (see Fig. S5). The variability of $NO_X$ reflects the variability of the traffic emissions, for stations within the influence of traffic, and this should be generally applicable throughout Europe. 72 sites are used for the calculation. The calculation of the distribution correction is carried out numerically after calculation of the $NO_2$ concentrations.

Implementation of the frequency distribution for concentrations has a significant impact, with a general reduction in $NO_2$ concentrations compared to the annual mean calculation using Eq. 1. This reduction leads to correction terms ($f_{no2,pdf}$) of between 0 to -25%. The highest corrections occur around $NO_X = O_X$, where Eq. (1) shows the most non-linear behaviour. On average for all station sites in Europe, around a -16% reduction on the initially calculated $[NO_2]_{mean}$ has been determined. In contrast to the distribution correction for concentrations, the distribution correction for $J$ leads to an increase in $NO_2$ of around 6%. This is because roughly half of the frequency distribution for $J$ is 0, i.e night time, when there is no photo-dissociation. In Sect. 5.5 the impact of this and other chemistry schemes is further discussed. More information concerning this scheme is contained in the supplementary material S1.4.

## 3 Example maps

In this section we present example maps that are generated from the EMEP model and uEMEP simulations. 100 km example tiles are shown in Fig. 3 - 5, demonstrating the original EMEP model calculations and the downscaled maps using uEMEP for $NO_2$, $O_3$, and $PM_{2.5}$. The uEMEP calculations are made on an x-y projected map commonly used for European mapping. The projection used is the European ETRS89-LAEA projection (EPSG: 3035). Maps presented are shown on latitude and longitude which means that the projected uEMEP tiled maps do not necessarily follow the North-South direction.

The downscaled maps resolve more variability between stations. Compared with the EMEP model maps, uEMEP maps have higher concentrations of $NO_2$ and $PM_{2.5}$, and lower concentration of $O_3$, in heavy traffic and populated areas due to the high-resolution proxy dataset.

## 4 Validation

Observed annual mean concentrations of $NO_2$, $PM_{2.5}$, $PM_{10}$, and $O_3$ from Airbase (European Environment Agency, 2018) are used for comparison with both the EMEP model and uEMEP calculations. All valid Airbase stations with more than 75% coverage are used in the validation and are assumed to be sited at 3 m above the surface. Results for the year 2018 are presented.





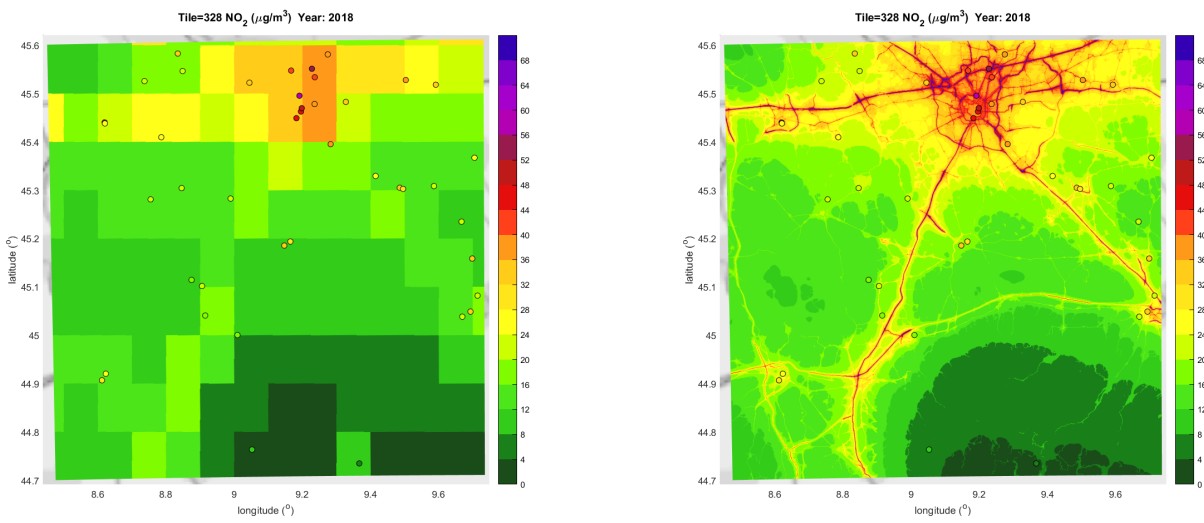

**Figure 3.** Calculated $NO_2$ concentrations in the 100 km tile (nr. 328) for 2018, part of the all European calculation at 100 m resolution. Left the EMEP model calculation at $0.1^o$ and right the uEMEP calculation at 100 m resolution. The city in this tile is Milan. Airbase stations are shown as circles.

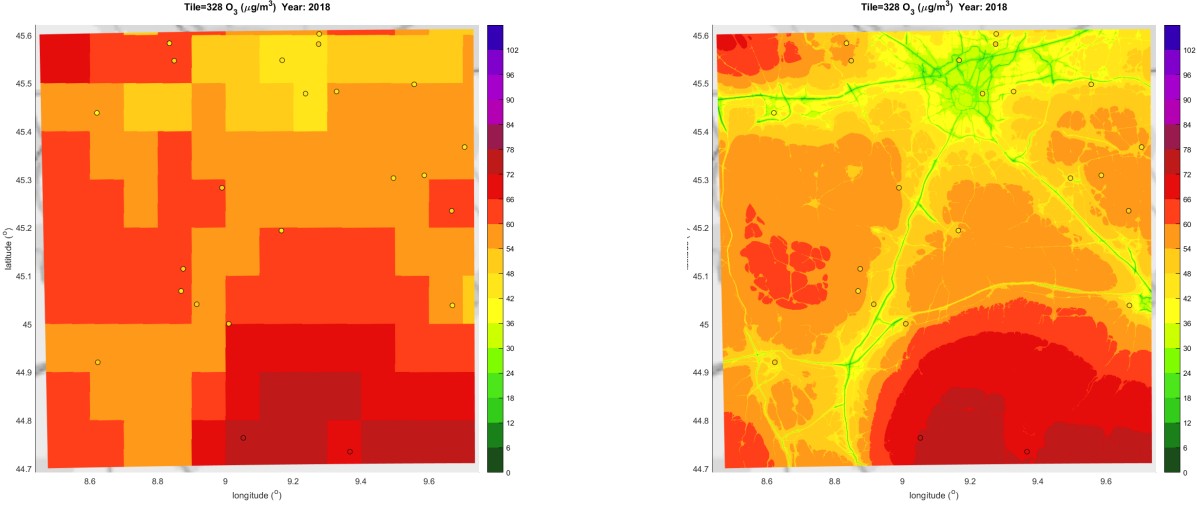

**Figure 4.** Calculated $O_3$ concentrations in the 100 km tile (nr. 328) for 2018, part of the all European calculation at 100 m resolution. Left the EMEP model calculation at $0.1^o$ and right the uEMEP calculation at 100 m resolution. The city in this tile is Milan. Airbase stations are shown as circles. This is the same tile as is shown in Fig. 3.





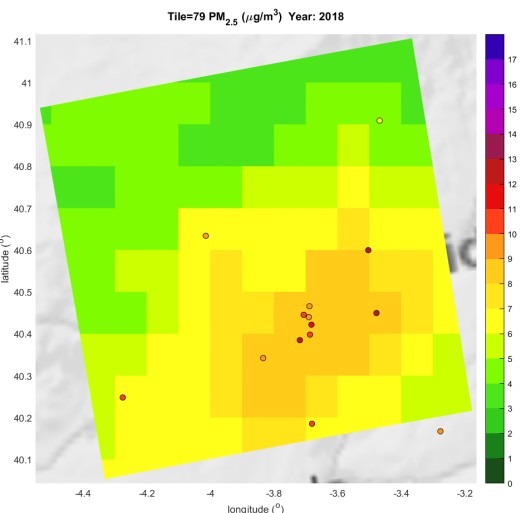
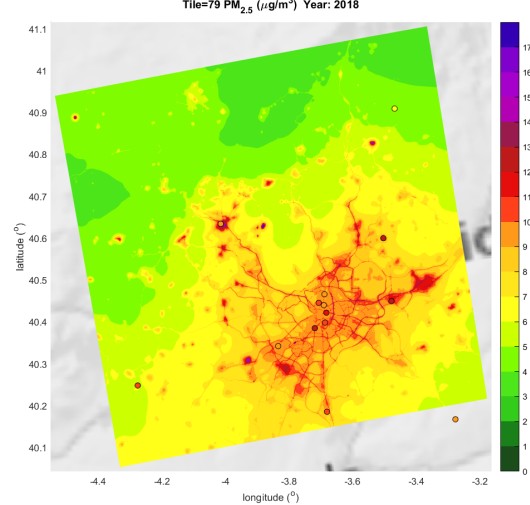

**Figure 5.** Calculated $PM_{2.5}$ concentrations in the 100 km tile (nr. 79) for 2018, part of the all European calculation at 100 m resolution. Left the EMEP model calculation at $0.1^o$ and right the uEMEP calculation at 100 m resolution. The city in this tile is Madrid. Airbase stations are shown as circles.

Results focus on the spatial correlation ($r^2$) and on the relative bias (Bias). For station sites the downscaling with uEMEP is performed on 25 m sub-grids, which is of sufficient resolution to spatially represent traffic sites. However, since the Gaussian model used does not take into account buildings or obstacles, then traffic sites in street canyons or built up areas may be poorly represented.

## 4.1   NO$_2$

In Fig. S6 and Fig. S7 scatter plots for NO$_2$ are shown for each country and Europe as a whole. These results are summarised in Fig. 6 where the annual mean concentration and spatial correlation are shown.

In a majority of countries the spatial correlation for NO$_2$ is more than doubled when implementing uEMEP. The two exceptions are Ireland (IE), where the spatial correlation hardly changes with the downscaling, and Bosnia and Herzegovina (BA), where the correlation is significantly reduced. Both these countries have very few stations. The highest spatial correlation is for Poland (PL) with $r^2 = 0.85$.

It is worth noting that the average spatial correlation per country is $r^2 = 0.62$ which is higher than the spatial correlation when assessed for all stations in Europe ($r^2 = 0.57$). This indicates that a part of the variability occurs between countries and can be interpreted to reflect differences related to emission reporting from each country. If the NO$_X$ emissions from individual countries have uncorrelated bias then this will reduce the overall spatial correlation.





The relative bias is improved for all countries with the exception of Greece (EL), which is the only country with a significant
positive bias. Overall for Europe bias is improved from -46% for the EMEP model to -18% when using uEMEP. Of the 28
countries with 10 or more monitoring sites, 18 of these have an absolute bias less than 25% after downscaling. Turkey (TR)
has the largest negative bias, after downscaling, of -45%.

## 4.2    PM$_{2.5}$

In Fig. S8 and Fig. S9 scatter plots for PM$_{2.5}$ are shown for each country and Europe as a whole. These results are summarised
in Fig. 7 where the annual mean concentration and spatial correlation are presented.

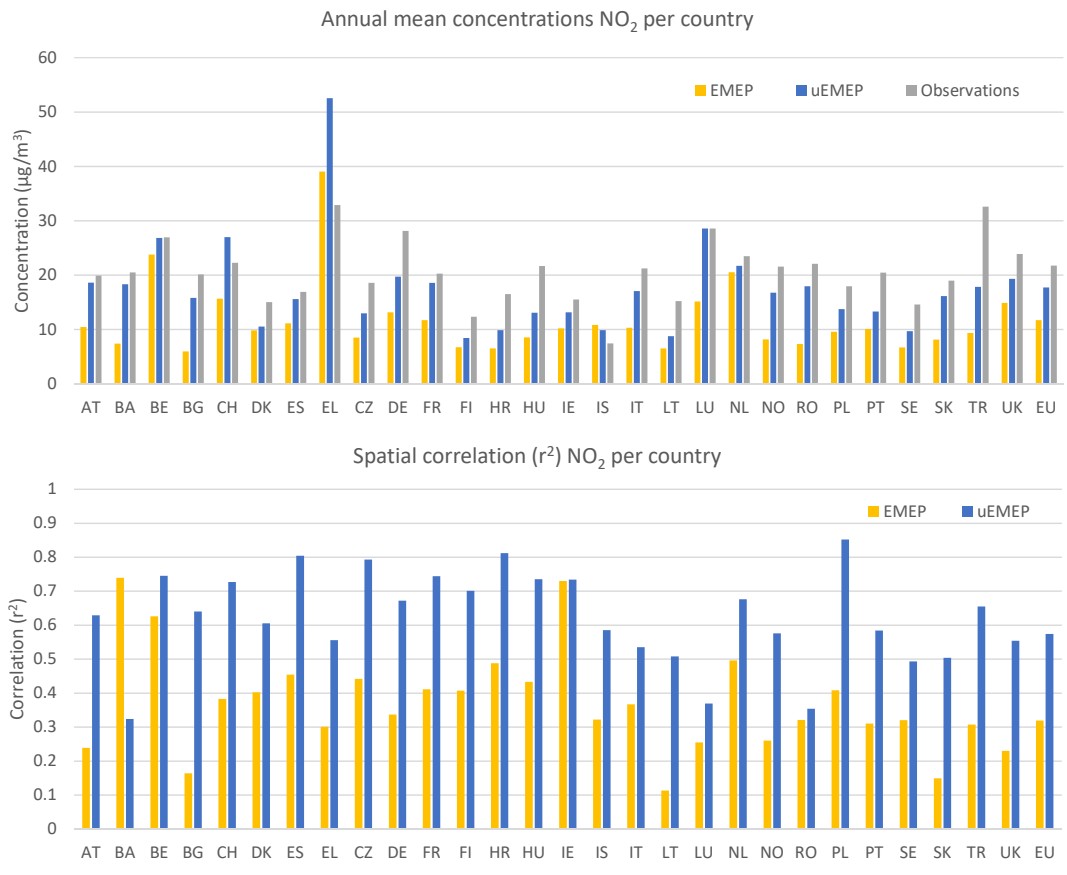

**Figure 6.** Annual mean NO$_2$ concentrations and spatial correlation ($r^2$) per country for 2018 calculated with the EMEP model and uEMEP
compared to Airbase observations. Only countries with 10 or more stations are shown but all stations are included in the final EU result. 3313
stations are included in the comparison.



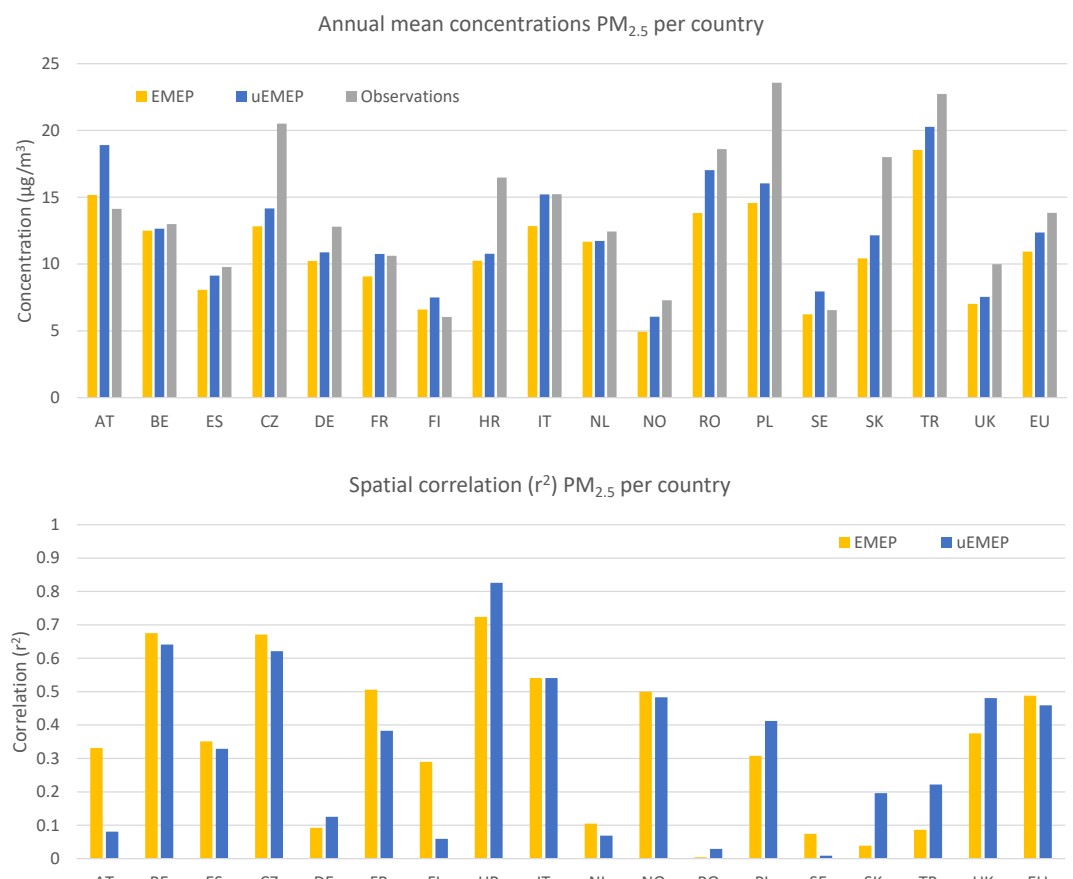

**Figure 7.** Annual mean $PM_{2.5}$ concentrations and spatial correlation ($r^2$) per country for 2018 calculated with the EMEP model and uEMEP compared to Airbase observations including all types of stations. Only countries with 10 or more stations are shown but all stations are included in the final EU result. 1376 stations are included in the comparison.

Unlike the $NO_2$ downscaling, there is generally no improvement in the spatial correlation when applying uEMEP for $PM_{2.5}$. Only 6 out of 17 countries show improved spatial correlation and overall for Europe there is a slight decrease, from $r^2 = 0.49$ for the EMEP model to 0.46 for uEMEP. This result is further discussed in Sect. 6.

The relative bias is however reduced for almost all countries. For Europe as a whole the relative bias went from -21% for the EMEP model to -11% for uEMEP. Only the three countries Austria (AT), Sweden (SE) and Finland (FI), that had almost no bias with the the EMEP model calculation, achieve a positive bias with uEMEP.

### 4.3  $PM_{10}$

In Fig. S10 and Fig. S11 scatter plots for $PM_{10}$ are shown for each country and Europe as a whole. These are summarised in Fig. 8.

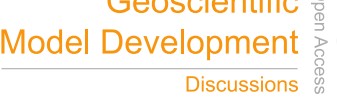



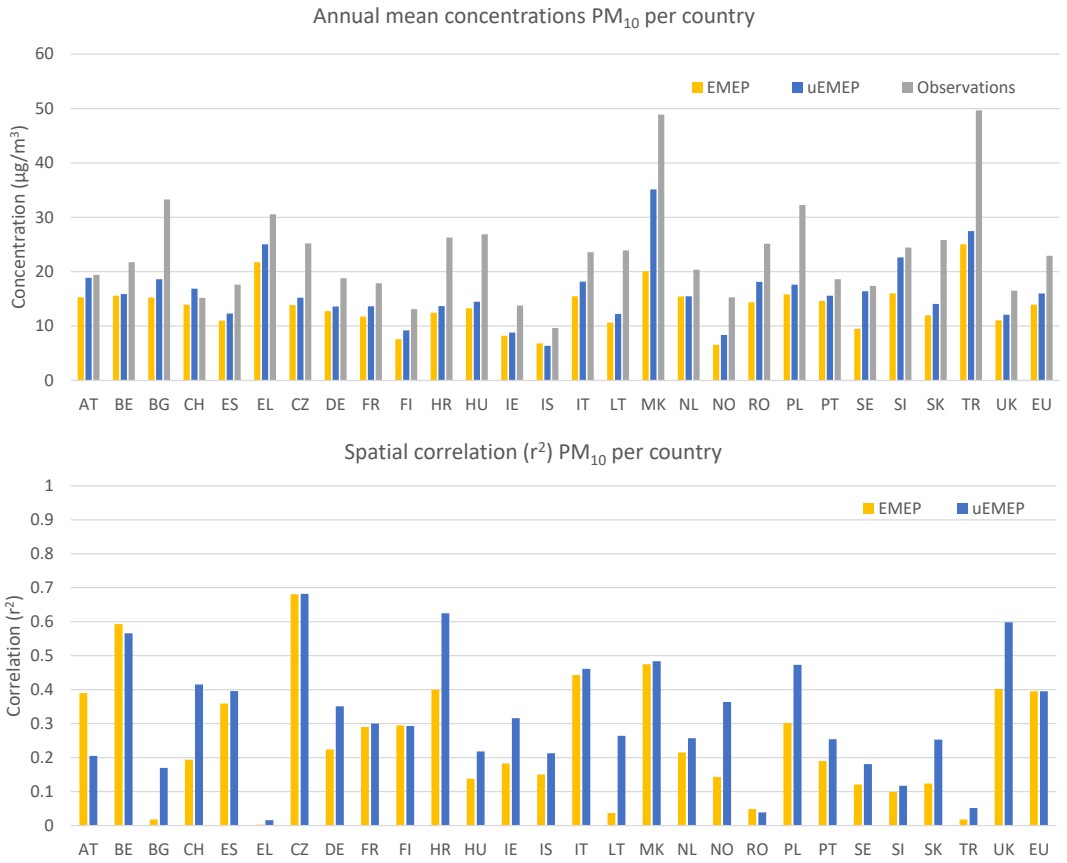

**Figure 8.** Annual mean $PM_{10}$ concentrations and spatial correlation ($r^2$) per country for 2018 calculated with the EMEP model and uEMEP compared to Airbase observations including all types of stations. Only countries with 10 or more stations are shown but all stations are included in the final EU result. 2891 stations are included in the comparison.

The results for $PM_{10}$ are similar to those for $PM_{2.5}$. In this case though the majority of countries, 21 out of 27, have improved spatial correlation with the application of uEMEP. The spatial correlation for all of Europe using uEMEP is unaltered compared to the EMEP model calculation, with $r^2$=0.34. This is lower than the spatial correlation found for $PM_{2.5}$ by around 0.12.

     As with $PM_{2.5}$ the relative bias is reduced with the uEMEP downscaling. For Europe we see the relative bias went from -39% for the EMEP model to -30% for uEMEP.

**4.4   $O_3$**

In Fig. S12 and Fig. S13 scatter plots for $O_3$ are shown for each country and Europe as a whole. These are summarised in Fig. 9.



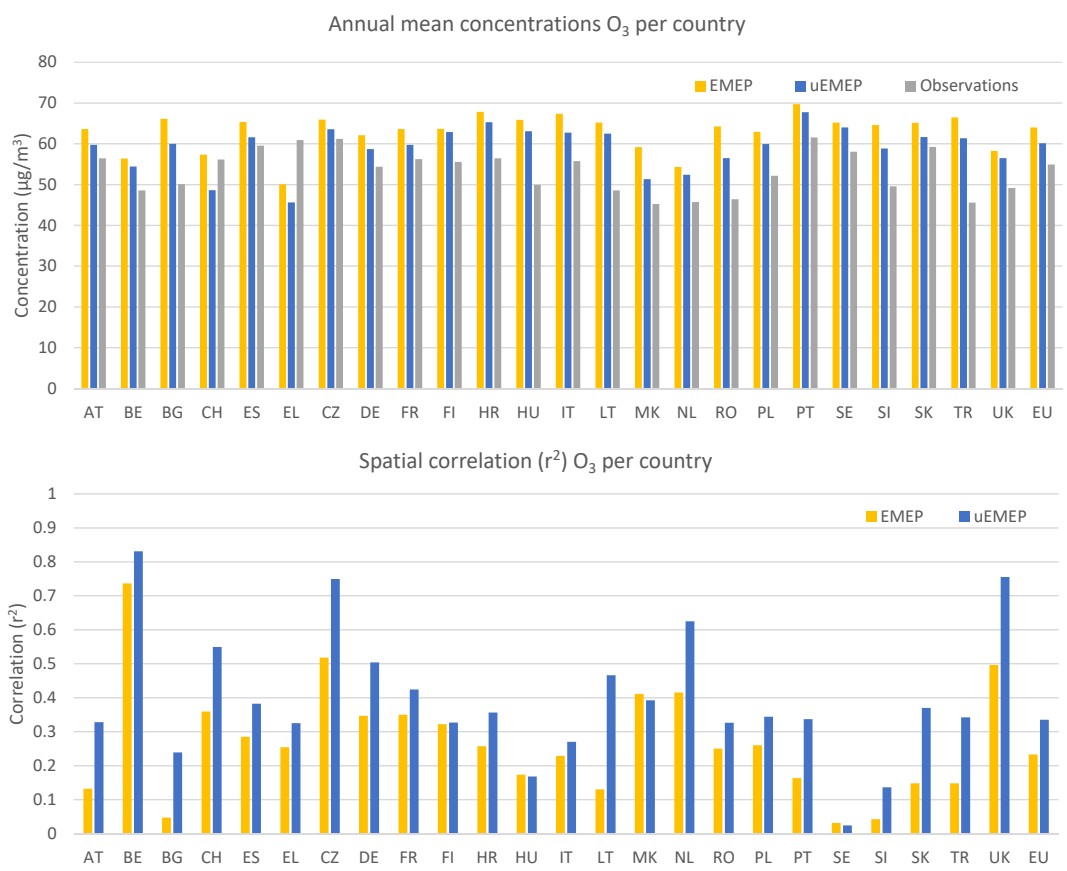

**Figure 9.** Annual mean $O_3$ concentrations and spatial correlation ($r^2$) per country for 2018 calculated with the EMEP model and uEMEP compared to Airbase observations including all types of stations. Only countries with 10 or more stations are shown but all stations are included in the final EU result. 1974 stations are included in the comparison.

Ozone is generally reduced with the downscaling due to an increase in $NO_X$ concentrations. In general for Europe we see a reduced positive bias from +16% for the EMEP model to +11% for uEMEP. Spatial correlation is also improved in 21 of the 210 24 countries. The only countries to show significant degradation in the downscaling results are Switzerland (CH) and Greece (EL). This is likely due to the overestimated $NO_X$ concentrations there (Fig. 6).

## 5 Sensitivity studies

In this Section we present the results of several sensitivity calculations using uEMEP. These include sensitivity to sub-grid resolution, traffic emission proxies, residential combustion emission proxies, sensitivity to alternative bottom up emissions in 215 Norway and sensitivity to the $NO_2$ chemistry scheme.




## 5.1  Sensitivity to resolution

When calculating at station positions a grid resolution of 25 m is used. However, when mapping all of Europe a lower resolution of 100 m is employed. In Fig. 10 we show the results of a change in resolution on the annual mean $NO_2$ and $PM_{2.5}$ concentrations for resolutions from 25 to 500 m.

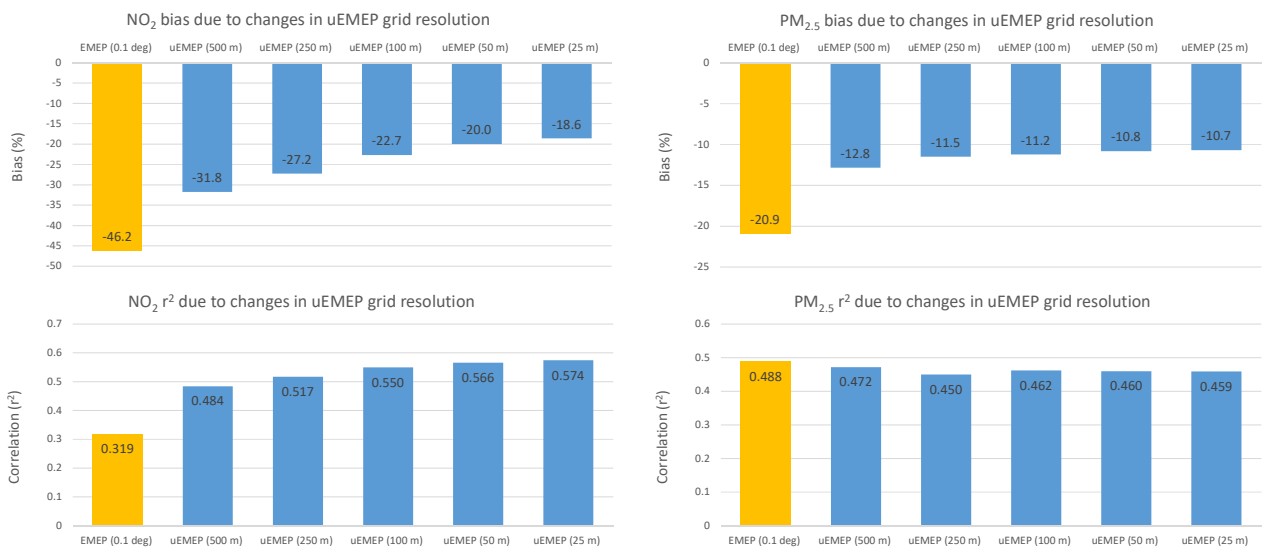

**Figure 10.** Change in bias and correlation as a result of changes in uEMEP resolution for the European calculations. Shown are the results for $NO_2$ and $PM_{2.5}$. Also included is the EMEP model $0.1^o$ calculation in yellow.

For $NO_2$ both bias and correlation improve with increasing resolution. 100 m calculations are on average 4% lower than the 25 m calculations. For $PM_{2.5}$ there is little change in bias between the different resolutions. Both shipping and residential combustion sources are only provided at 250 m so any further change in model results at lower resolutions will be due to the traffic contribution only. Spatial correlation is basically unchanged for $PM_{2.5}$ at all resolutions.

## 5.2  Sensitivity to OSM weighting

In Fig. 2 the weighting imposed on the OSM road categories is shown. This weighting specifies the relative contribution of the different road categories to the redistribution of the gridded traffic emissions in uEMEP. This weighting is based on an analysis of Norwegian traffic data but it is worthwhile assessing the sensitivity of the calculated $NO_2$ concentrations using different weights. To assess this sensitivity a power law is applied to the weighting. For power indices greater than 1 then more weighting is applied to the the major roads, for power law indices less than 1 then more weight is applied to the the minor roads.

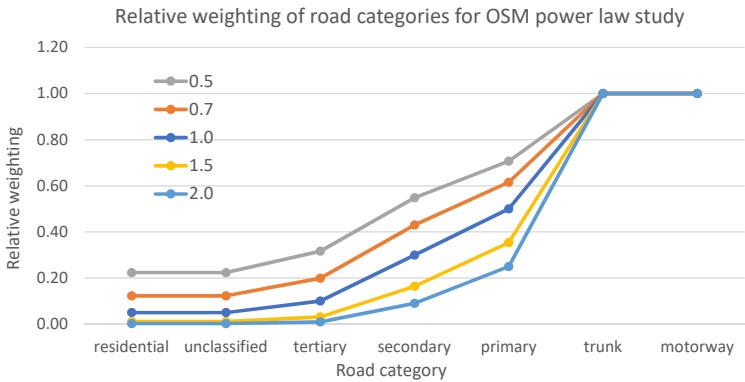

**Figure 11.** OSM weighting, relative to Motorways, that results with a change in the power law index when applied to the initial weights.

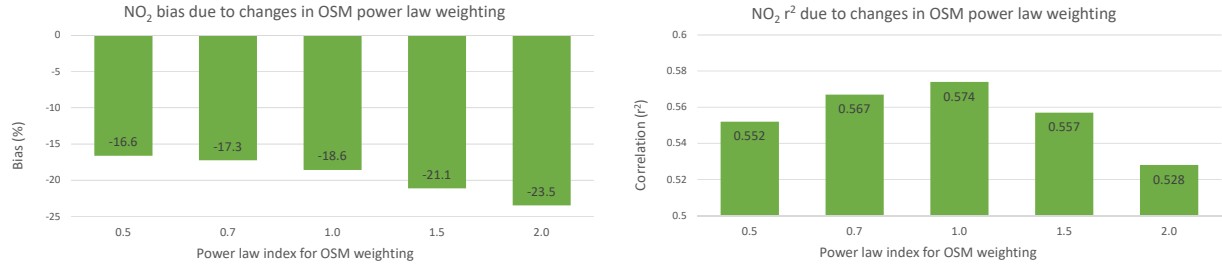

**Figure 12.** Change in bias and correlation for $NO_2$ as a result of changes in the power law index applied to the OSM road traffic weighting. Lower power law indices give more weight to minor roads, higher power law indices give more weight to major roads.

The different weights for the different power law indices are shown in Fig. 11. The results of this sensitivity test, presented in terms of relative bias and spatial correlation ($r^2$), are shown in Fig. 12.

Bias is quite strongly affected by the change in weighting. Higher concentrations are calculated when more weight is given to the minor roads. This is likely because most measurement sites are not on major roads. Increasing the weighting to minor roads will generally increase the urban background levels. The spatial correlation is highest for the current weighting with a

power index of 1. This confirms that the initial estimate, based on Norwegian traffic, reflects a more general distribution of traffic in Europe. If real traffic volume were available then the weighting would be more precise. Tests on Norwegian data, Sect. 5.4, confirm that spatial correlation is significantly improved when using real traffic data for the redistribution weighting.





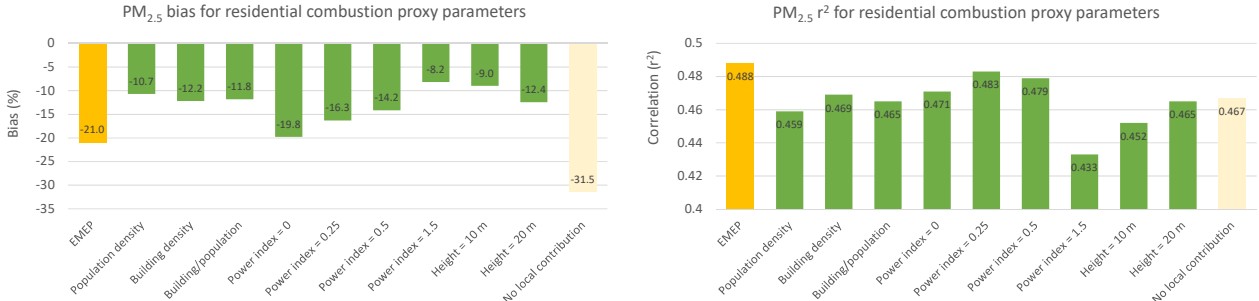

**Figure 13.** Change in bias and correlation for PM$_{2.5}$ as a result of changes in the residential combustion proxy. A lower power law index gives less weight to the population redistribution, a higher power law indices give more weight. 'Building/population' is the building density masked by population data. See text for details.

## 5.3 Sensitivity to the residential combustion emission proxy

For the PM$_{2.5}$ calculations presented in Sect. 4.2 population density data at $0.0025^o$ has been used to redistribute the residential
combustion emissions in uEMEP. The results indicate a slightly reduced spatial correlation but also with an improved negative
bias. In this section we assess the sensitivity of the redistribution proxy to a number of alternative proxies. Firstly a power law
is applied to the population density data. A lower power law index will reduce the weighting towards highly populated regions.
A power law index of 0 will work as a mask, redistributing the EMEP emissions evenly to any 250 m sub-grid that contains
population. As an alternative to the population data, building density data has also been extracted from the OpenStreetMap
dataset. This has also been placed on a $0.0025^o$ grid for all of Europe. Two alternatives with this proxy are tested. The first
using building density as the weighting proxy and the second using building density masked with population, so that only areas
with both buildings and population are used for redistribution. In addition to the alternative proxy data the sensitivity of the
calculations to emission height, currently set to 15 m, is also assessed.

    The results are shown in Fig. 13. Here we see that a power law of 0.25 gives slightly improved spatial correlation and that
the use of building density also slightly improves correlation compared to population. However, none of the alternative proxies
significantly improves the spatial distribution of PM$_{2.5}$ and none attain the spatial correlation of the EMEP model calculations
at $0.1^o$. There is a general trend for reduced negative bias to lead to reduced spatial correlation in all calculations, so when the
contribution from the downscaled residential combustion increases then correlation reduces. This infers that the redistribution
is not improving the results.

In addition to the proxy sensitivity the result of the EMEP model calculation where all local EMEP model grid contributions
($\pm 1^o$) have been removed is shown in Fig. 13. This shows firstly that around 10% of the PM$_{2.5}$ in the EMEP model comes
from within this local region and that the inclusion of these emissions does add to improved spatial correlation at the EMEP
model $0.1^o$ scale, from $r^2 = 0.467$ to $0.488$. Here we see more clearly that while the bias is improved by downscaling the spatial





correlation is not and is similar to the correlation obtained from the non-local contributions. However, it is possible to achieve
improved spatial correlation when more appropriate downscaling proxies are used. This is presented in Sect. 5.4.

### 5.4 Results of improved emission data in Norway

Throughout the uEMEP downscaling simulations we used the $0.1^o$ country reported emission data and redistributed it using
population, OpenStreetMap data and AIS shipping data as redistribution proxies. However, many countries have more detailed
emission data sets, including Norway, that could be used to improve the downscaling calculations. To test the impact of more
realistic spatial distributions of emissions, the emission and emission proxy data used in Norway are replaced in the EMEP
model and uEMEP calculations with the emission data currently used in the national air quality forecasting in Norway. Details
surrounding these emissions can be found in Denby et al. (2020) and Grythe et al. (2019). The most important differences
between the Norwegian and European emissions and emission proxy data are: (1) Traffic volume data from the Norwegian
national road database is used instead of OSM weighting. Exhaust emissions are based on emission factors using a bottom
up methodology and $NO_X$ emissions are additionally corrected for temperature. (2) The non-exhaust road dust emissions are
calculated with the NORTRIP model (Denby et al., 2013a; Denby et al., 2013b) which are significantly larger than the current
national estimates reported for Norway. (3) The total Norwegian residential heating emissions of PM are the same for both the
Norwegian and the European emissions but the Norwegian emissions have been redistributed using the MetVed model (Grythe
et al., 2019), which uses much more detailed information than just population to distribute the residential heating emissions at
250 m. (4) The Norwegian emissions and the uEMEP proxy data are entirely consistent with each other since the Norwegian
emissions are aggregated grid emissions based on the fine scale emission data.

We make four separate downscaling calculations for Norway using the two emissions, 'European emissions' and 'Norwegian
emissions', and the two high-resolution proxy datasets, 'European proxy downscaling' and 'Norwegian proxy downscaling',
respectively. Shipping is not changed in these simulations and in this case the calculation year is 2017. Though the resolution of
the EMEP model in the Norwegian forecasting system is nominally 2.5 km, for these simulations we use the same $0.1^o$ EMEP
model grid resolution. The results are shown in Fig. 14 for $NO_2$, $PM_{2.5}$ and $PM_{10}$ where the relative bias (%) and correlation
($r^2$) are presented.

For $NO_2$ in Norway the large negative bias seen in the EMEP model is almost completely removed by the use of the traffic
downscaling, using either the Norwegian or European emission data. On a national level the local Norwegian (bottom up) traffic
$NO_X$ emissions are roughly 25% higher than the EMEP (top down) emissions. $NO_2$ concentrations are slightly overestimated
when using the Norwegian proxy data for traffic. Spatial correlation is improved with the use of the Norwegian proxy data
for traffic, compared to European emissions that use OSM data, from $r^2$ = 0.6 to 0.72. It is worth noting that in the complete
Norwegian calculation reported in Denby et al. (2020) using hourly calculations that the spatial correlation is even higher at $r^2$
= 0.78, but the bias is less at -5%.

For $PM_{10}$ both bias and correlation are significantly improved with the implementation of the local emissions and proxies.
This is to a large extent due to the improvement in the road dust emission contribution but also due to an improvement in the
residential heating distribution. Spatial correlation is also significantly increased, from $r^2$ = 0.27 to 0.49.

**Figure 14.** Change in bias and correlation as a result of changes in Norwegian emission and emission proxy data for $NO_2$, $PM_{2.5}$ and $PM_{10}$ calculations. 'European emissions' are the emissions used for all of Europe and 'Norwegian emissions' replaces these emissions for traffic and residential heating with alternative emissions used in the Norwegian air quality forecasting system. 'European' and 'Norwegian' proxy downscaling are explained in the text. Calculation year is 2017. The number of available stations is 41, 36 and 44 respectively for $NO_2$, $PM_{2.5}$ and $PM_{10}$.

For $PM_{2.5}$ biases are very similar for both the European and Norwegian proxy data sets when using either the European or Norwegian emissions. The spatial correlation however is significantly higher when using the Norwegian emissions, both at grid level and after downscaling. There is significant improvement, $r^2$ increases from 0.37 to 0.55, when both changing European emission to Norwegian emission and changing the residential heating proxy from population (European proxy) to the MetVed model (Norwegian proxy). This indicates that improved spatial representation can be attained when both the gridded and the proxy data are consistent and more representative. However, little can be improved with downscaling when the initial gridded





emissions are not well distributed, even with improved proxy data. Interestingly we see the same result as reported in Sect.
4.2, that the spatial correlation is reduced when applying the European proxy data to the European emissions. These results
indicate that significant improvements can still be obtained in the downscaling if improved emissions and emission proxies are
implemented.

### 5.5   Sensitivity to the $NO_2$ chemistry scheme

Included in uEMEP are a number of simplified $NO_2$ chemistry schemes, used to derive downscaled $NO_2$ concentrations from
$NO_X$ and $O_3$ concentrations. In the results presented so far we have used the weighted travel time parcel method, as applied
and described in Denby et al. (2020), with the additional use of the frequency distribution correction described in Sect. 2.4.
Two additional chemistry-based schemes and two empirically-based schemes are also available. The two alternative chemistry
schemes are the photo-stationary formulation and an alternative stationary formulation that also allows for deviation from the
photo-stationary state (Maiheu et al., 2017). The first empirical scheme is the Romberg scheme (Romberg et al., 1996), also
described in Denby et al. (2020), that directly converts $NO_X$ to $NO_2$ concentrations. The parameters for this equation have
been updated by fitting to all available Airbase data for the year 2017. The other empirical formulation is the SRM scheme
(Wesseling and van Velze, 2014) that is also based on a fit to measurement data but includes background $O_3$ as one of the input
parameters. The advantage of the two empirical fits is that they should convert $NO_X$ to $NO_2$ in a manner that is consistent with
the observations, and as such can be applied to annual mean concentrations directly, without any correction for non-linearity.
All five methods are described in the Supplementary material, Sect. S1.

In Fig. 15 we provide the results of the sensitivity tests, showing bias and correlation for both $NO_2$ and $O_3$. The three
chemistry based schemes give similar results indicating that in all three cases the calculations are close to photo-stationary.
The two empirical fits also give similar results, with the largest negative bias in $NO_2$ given by the Romberg scheme with
-25%. Since the Romberg scheme is specifically designed to reflect measurements, providing the correct $NO_2/NO_X$ ratio, this
means that the chemistry schemes are overestimating the $NO_2$ contribution when applied to annual mean concentrations. This
is partially due to the positive bias in the EMEP model $O_3$ concentrations of 16%, but this only accounts for around 4% of the
additional $NO_2$. Included in Fig. 15 is the annual mean calculation without the frequency distribution correction ("Travel time
(annual)"), showing a 10% difference in bias when compared to calculations that use this correction. Spatial correlation is also
improved by using the frequency distribution methodology.

### 6   Discussion

Downscaling only applies to emissions within a limited region of $\pm 0.1^o$ surrounding each receptor sub-grid. Based on the
uEMEP calculation, the local contributions to $NO_X$ are significantly larger than for PM. The different source contributions at
measurement sites are given in Table 2 and this shows that, on average in Europe, 58% of the $NO_X$ contributions come from
traffic within this limited region. In contrast only 19% of the $PM_{2.5}$ is attributable to residential heating, the largest downscaled
contribution, from inside this region.



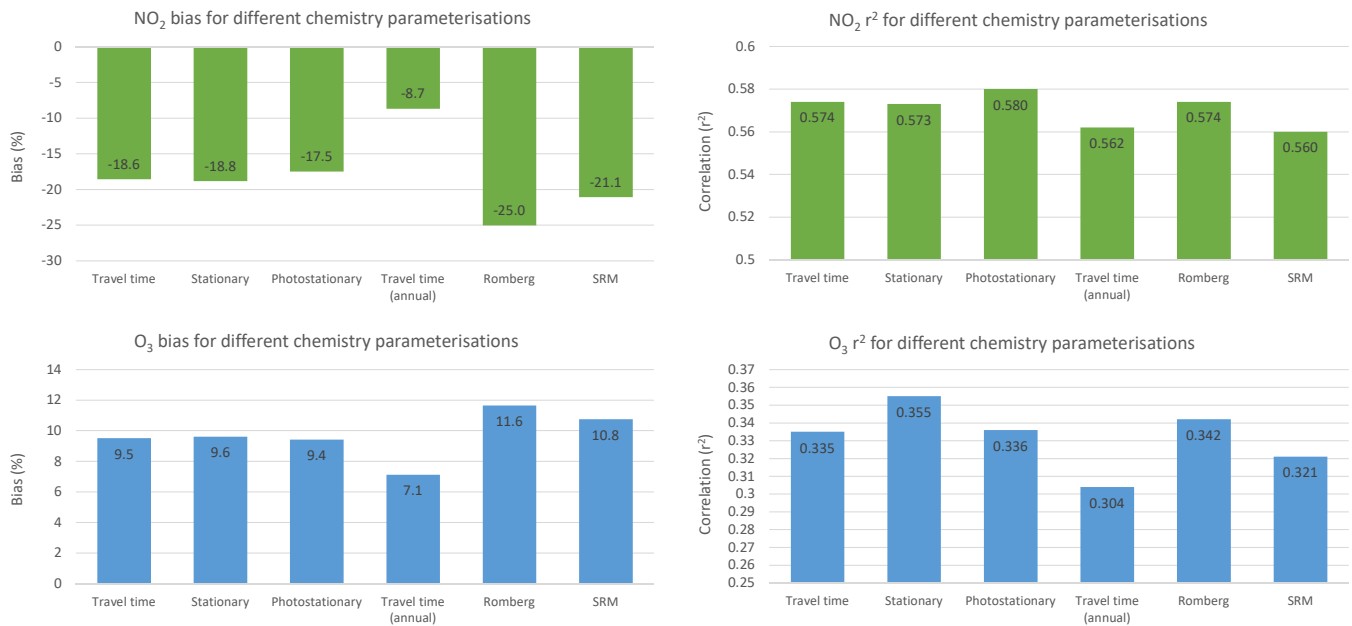

**Figure 15.** Change in bias and correlation for $NO_2$ and $O_3$ with implementation of 6 different versions of the chemistry schemes. See text for details.

**Table 2.** Source contribution to all air quality stations in Europe calculated with uEMEP. uEMEP local contributions are from emissions within an region of $\pm\ 0.1^o$ in both latitude and longitude. Non-local EMEP model contributions are all emissions from outside this region for the downscaled sources as well as other sources within this region that are not downscaled.

| Source | $NO_X$ ($\mu g/m^3$) | $PM_{2.5}$ ($\mu g/m^3$) | $PM_{10}$ ($\mu g/m^3$) |
|---|---|---|---|
| Traffic (GNFR6) | 13.9 (58%) | 0.71 (6%) | 1.1 (7%) |
| Residential heating (GNFR3) | 1.8 (8%) | 2.2 (19%) | 2.6 (16%) |
| Shipping (GNFR7) | 0.30 (1%) | 0.01 (0.1%) | 0.01 (0.1%) |
| Non-local EMEP | 7.9 (33%) | 8.4 (75%) | 12.3 (77%) |
| Total | 23.9 (100%) | 11.3 (100%) | 16.0 (100%) |

$NO_2$ is well modelled with high spatial correlation for many countries, but still with a significant negative bias of -18%. There is significant variation in bias between countries even though the methodology is consistently applied to all countries. This may be attributable to the various methods used for generation of the national emissions. Though the problem remains that uEMEP does not take into account dispersion in street canyons, where a number of traffic site measurements are made, it is generally the case that the spatial representativeness of the uEMEP calculations is suitable for comparison with these measurements.





Variation in bias between countries is then no longer a case of a mismatch in resolution but most likely reflects bias in the national emissions. uEMEP may be used to investigate this variability between countries further and to help harmonise future emission inventories across Europe.

There is a significant difference between the results achieved for the downscaling of PM compared to $NO_2$. $NO_2$ is dominated

by traffic emissions and this is spatially well defined using OSM as a proxy. The largest contributor to PM is residential heating which uses population as a downscaling proxy, so it appears that this is not a good proxy for high-resolution emission redistribution. Though clearly residential heating emissions occur where people live there can be large variation from city to city and from urban to suburban and to rural areas as heating practices vary significantly depending on housing type and on availability of alternative heating sources. To some extent this has been taken into account in the emission inventory at $0.1^o$,

but the emission proxy used in uEMEP is likely not consistent with the EMEP emission inventory.

The Norwegian sensitivity tests show that when consistent emissions and emission proxies are used then spatial correlation can be significantly improved. For the application of uEMEP in Europe this was not the case since each country has their own methodology for calculating gridded EMEP emissions that may or may not make use of the downscaling proxies applied in uEMEP. A more consistent approach, as applied in Norway, would be to use the same spatial redistribution proxies in both

the gridded EMEP emissions and the downscaling proxies. This would require additional interaction and cooperation between emission inventory developers and air quality modellers.

It is worth noting that no selection of the Airbase monitoring data was carried out. All available stations with more than 75% coverage were used. This includes mountain stations, all traffic stations as well as industrial sited stations. In comparisons with the EMEP model these types of sites are often removed. All stations were also assumed to be sited at 3 m above the surface.

It is quite possible that different results would be obtained if a selection of stations was carried out. This will be assessed at a later time.

## 7 Conclusions

Downscaling of annual mean concentrations from the EMEP model have been carried out for $NO_2$, $PM_{2.5}$, $PM_{10}$, and $O_3$ using the uEMEP model. Downscaling redistributes EMEP gridded emission data, using suitable proxy data, to high-resolution sub-

grids and then calculates the sub-grid concentrations using a Gaussian dispersion model. These are then recombined with the EMEP model concentrations in a consistent way that avoids double counting of the emissions. Maps for all of Europe have been produced at a resolution of 100 m and concentrations at all Airbase measurement sites have been calculated at 25 m.

The results for $NO_2$ show significant improvement with a doubling of spatial correlation for most countries and a significant reduction in negative bias. For $NO_2$ the downscaling works very well, which is due to the fact that $NO_X$ emissions are mainly

attributable to traffic and these emissions are well defined spatially with the proxy data used. $O_3$ concentrations are decreased due to higher $NO_X$ concentrations. Both concentrations and spatial correlations of $O_3$ are better simulated with uEMEP.

Neither $PM_{2.5}$ nor $PM_{10}$ shows any improvement in spatial correlation with the downscaling, though the negative bias in PM concentrations is improved. The spatial distribution of PM emissions can be improved, as demonstrated for Norway, with





more accurate proxy data, but emissions of PM remain difficult to quantify properly at high resolutions and will require further
effort. One way forward is to harmonise the proxies used for both the EMEP gridded emissions and the uEMEP downscaling.
This has been shown to improve results in Norway.

   Downscaling can provide additional information concerning the contributions of local sources. This may be combined with
the EMEP model source-receptor calculations to provide a more complete picture of local and long-transported contributions.
The method can lead to a better assessment of local verses regional mitigation strategies to improve air quality in Europe at
high resolution. It also shows good potential to be used to improve exposure estimates.

*Code and data availability.* The uEMEP_v6 model used in this study is archived on Zenodo (https://doi.org/10.5281/zenodo.4923185), as
are MATLAB scripts for visualisation of European uEMEP calculations presented in this paper (https://doi.org/10.5281/zenodo.4923224).
The latest development of uEMEP can be found at https://github.com/metno/uEMEP.

*Author contributions.* BRD developed the downscaling method of Europe in uEMEP and wrote most of the text. QM carried out the EMEP
model calculations, processed the raw proxy data as input files for uEMEP, and contributed to the text. EGW provided the frequency distri-
bution correction and contributed to the text. HF internally reviewed and contributed to the text.

*Competing interests.* The authors declare that they have no conflict of interest.

*Acknowledgements.* uEMEP development was supported by the Research Council of Norway (NFR), grant no. 267734, the Norwegian
Public Roads Administration (Statens Vegvesen), the Norwegian Environment Agency (Miljødirektoratet) and the Ministry of Climate and
Environment (Klima-og miljødepartementet). We thank Dr. David Simpson for a nice internal review.





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
