# Peer review of "Downscaling of air pollutants in Europe using uEMEP\_v6"

_Geoscientific Model Development, 2021_

## Author Comment (AC1)

**Author response to Referee Comments and Editor Comment**

Many thanks to Dr. David Topping and the anonymous referees for the constructive comments. The manuscript is definitely improved. Below is our response to the RC1, EC1, and RC2, respectively. The original comment is in black, while our response is in blue. The added/modified text is shown in italic. The reference list is at the end of the document. Note that we've added one new figure as Fig. 1, so the figure number in the revised manuscript equals to the figure number in this response plus 1.

**(RC1)**

General comments

This preprint presents high-resolution air quality modelling at local scale/street level across Europe, performed with the uEMEP_v6 model, which downscales emission input and combines regional calculations with the EMEP model with Gaussian plume modelling of receptor points to obtain annual mean concentrations of $NO_2$, $PM_{2.5}$, $PM_{10}$ and $O_3$ in a very high resolution subgrid (down to 100 m resolution). Results presented are comparisons of EMEP and uEMEP model results country by country for Airbase monitoring stations across Europe, as well as sensitivity studies with respect to resolution, weighting of relative traffic distribution, proxies for residential combustion emissions, use of national emission and proxy data with higher detail than the EMEP data, and $NO_2$ chemistry schemes for the $NO_2$-$NO_x$-$O_3$ reactions. Model results are seen to significantly improve in comparison with measurements for $NO_2$ and $O_3$ when the uEMEP model is applied whereas little improvement for $PM_{2.5}$ and $PM_{10}$ is gained using the downscaling approach.

The manuscript addresses the highly relevant scientific question concerning how to obtain high resolution air quality estimates at the urban scale in an operational way, using publicly available data. The different parts of the methodology are known, but the combination is novel and the uEMEP model tool is potentially extremely useful for air quality and population exposure studies across Europe. The manuscript is well-written and the structure and argumentation of the study

is easy to follow with a few minor details that could be clarified (see Specific comments). All model code for the uEMEP model and the visualization tools is publicly available for this study.

Specific comments:

When investigating annual mean values, a rotation symmetric approach for the Gaussian dispersion is used. This implies the assumption that the wind is equally distributed from all directions over a year. In reality these wind distributions will probably be quite different. Could the authors elaborate on if this rotation symmetric approach is more or less accurate in different locations? And what potential comparisons of the approach to applying actual annual met-data would show?

The assumption is usually not met in reality, but the result of this simplified method agrees very well with the hourly calculation. In Sect. S5.1 of Denby et al. (2020), we compared the annual and hourly calculations for the same 100 x 100 $km^2$ region surrounding Oslo with a resolution of 100 m. Figure S2 and S3 show very similar results for the two model calculations. We have not assessed the error on a European wide basis and so cannot quantify this. From the tests carried out in Norway there is no indication that this approximation leads to errors larger than other uncertainties in the modelling.

When the OSM data are applied across Europe, a weighting based on the Norwegian traffic data is used – can the authors elaborate on what this means for the distribution of local emissions in the rest of Europe?

In Sect. 5.2, we tested other weighting scenarios by changing the power law index which is applied to the initial weights. Results show that the spatial correlation is highest for the initial weights based on Norwegian traffic, and bias is also reasonable. It indicates that weightings derived from Norwegian traffic can reflect a general distribution of traffic in Europe. Moreover, the results also indicate that higher weightings for the minor roads can improve the bias, as the urban background level is increased. This provides insights for the rest of Europe where local traffic data is available.

The NO-NO$_2$-O$_3$ chemistry for annual mean values is based on a calculation including a frequency distribution of the concentrations of NO, NO$_2$ and O$_3$. It is a little unclear from the manuscript, if these frequency distributions are acquired from Norwegian stations only, or if all available measurements from Europe have been taken into account? The dependency on solar input in the photochemical reactions must mean that the frequency distribution will differ across Europe?

The frequency distributions of NO$_X$ and O$_X$ are log-normally distributed. The standard deviation and mean values are derived from earlier modeling results at 72 Norwegian stations. Since the variability of NO$_X$ reflects the variability of the traffic emissions, this should be generally applicable throughout Europe for stations within the influence of traffic. The magnitude of the O$_X$ variability will likely be different across Europe, depending on O$_3$ levels but this distribution has less impact on the results than NO$_X$. Ideally this analysis should have been done for all European stations but this requires a sizable effort to assess hourly data from paired stations.

As Eq. (3) shows, the distribution of J follows the distribution of the solar zenith angle (ZA). This distribution is based on the latitude and longitude of each European grid point so is valid everywhere.

The results of the uEMEP calculations correspond to street-level, but the building configuration is not included, and common situations with development of street-canyon circulation vortices are therefore also not taken into account. Can the authors elaborate on what this means for the results at street-level in the large cities with tall and dense building mass?

There will likely be an underestimate at the street-canyon sites, since the Gaussian model does not consider obstacles and does not include any form of street canyon parameterisation. In Europe there are a large number of traffic sites in fairly open environments that do not require street canyon modelling. Exactly how many traffic sites are not known. Reporting to the EU Commission (e-reporting) on siting of traffic stations is generally quite incomplete. Metadata provided in e-reporting was analysed for the EU commission in 2018 (Tarrasón et al., 2021). This indicated that the majority of stations are not street canyon sites, though the exact number was unclear.

In a study by Lefebvre et al. (2013) modelling of $NO_2$ was carried out in Antwerp using both Gaussian and street canyon modelling for 49 stations, a mix of both open road and street canyon sites. On average over all sites the increase in concentration was around 4% with the use of the street canyon model. For individual street canyon sites (15 sites) this increase was 11%. If Antwerp can be interpreted as a representative European city then we can expect a similar underestimation for all sites in Europe.

The uEMEP model is intended for applications over country scales, and the level of detail required for street canyon modelling is not easily achievable. Without including obstacles, the increased model resolution (up to 25 m) allows the concentration gradients at roadside to be better described.

At the end of the first paragraph of Sect. 4, we've added: "...traffic sites in street canyons or built up areas may be *underestimated. A study by Lefebvre et al. (2013) in Antwerp, where both Gaussian and street canyon models were applied at 15 street canyon modelling sites, showed an average street canyon modelling increment of just 11% for $NO_2$. We include all available sites in this study because in Europe the majority of traffic sites appear not to be street canyons, though information on this is unclear (Tarrasón et al., 2021). Also, even without including obstacles, the increased model resolution (up to 25 m) allows the concentration gradients at roadside to be better described.*"

Regarding PM2.5: All annual mean concentrations for ($NO_2$ and) PM2.5 increases when the uEMEP downscaling is applied compared to the EMEP model results. In two countries (Austria and Finland), the EMEP model is already overestimating the PM2.5 concentration, and applying the uEMEP model only increases the overestimation. Wouldn't the authors expect that downscaling using proxies would give a more precise result for the distribution, and thereby a more accurate replica of what is observed? Or is there an underlying risk, that uEMEP increases the concentrations in general?

The downscaling proxies only redistribute the original EMEP emissions, so that the total emissions in EMEP and uEMEP are the same. Therefore, uEMEP does not always increase the concentrations in general, but can resolve more variability, i.e. higher concentrations in heavy

traffic and populated areas, and lower concentrations vice versa, as shown in the example maps Fig. 3-5. The concentrations at validation stations are in general higher for uEMEP, because emissions around those stations are higher after redistribution, and validation results show that uEMEP has done a better job than EMEP in most of the countries and EU as a whole. However, as the reviewer points out, uEMEP further overestimates $PM_{2.5}$ concentrations in two countries (Austria and Finland). This could be due to the uncertainty of the proxies in those countries or that the EMEP emissions themselves are overestimated in these countries. Recent analysis of the CAMS residential combustion emissions by TNO themselves have indicated, personal communication, that these emissions are indeed overestimated in Austria. We discuss more about the uncertainty of residential emission proxy in Sect. 5.3.

Figure 10: is this result for the whole of Europe, i.e. a mean of all countries?

The result is for the whole of Europe, which is averaged over all Airbase stations with >75% coverage. We've added an explanation into the caption of Fig. 10: "*The results are based on the European calculations and all available Airbase stations are included.*"

Figure 11 and 12: It would be good with a little more explanation in the Figure captions, e.g. a note whether this is all of Europe, or only Norway.

In the caption of Fig. 11, we've added that "The scenarios are tested in the European calculations." In the caption of Fig. 12, we've added that "*The results are based on the European calculations and all available Airbase stations are included.*"

Figure 12: the conclusion that the correlation is clearly highest for power law index 1 is putting much trust in the decimals of the correlations. As the numbers are 0.567 (~0.57), 0.574 (~0.57) and 0.557 (~0.56), one could wonder how much the third decimal of the correlation estimate is worth in terms of accuracy?

The reviewer is right that the correlations don't differ much, but we can still say the power law index 1 can reflect a good (though not clearly the best) general distribution of traffic in Europe. We rephrase the sentence like this: "The spatial correlation is *among* the highest for the current weighting with a power index of 1. This confirms that the initial estimate, based on Norwegian traffic, reflects a *good* general distribution of traffic in Europe."

In the discussion: "Though the problem remains that uEMEP does not take into account dispersion in street canyons, where a number of traffic site measurements are made, it is generally the case that the spatial representativeness of the uEMEP calculations is suitable for comparison with these measurements." How do the authors know that this is the case?

In a study by Lefebvre et al. (2013) modelling of $NO_2$ was carried out in Antwerp using both Gaussian and street canyon modelling for 49 stations, a mix of both open road and street canyon sites. On average over all sites the increase in concentration was around 4% with the use of the street canyon model. For individual street canyon sites (15 sites) this increase was 11%. If Antwerp can be interpreted as a representative European city then we can expect a similar underestimation for all sites in Europe. We've added this reference to the text "... suitable for comparison with these measurements *(Lefebvre et al, 2013)*."

Technical comments:

Figure 14: the order of the components is $NO_2$, $PM_{2.5}$ $PM_{10}$, but in the text the order of discussion is $NO_2$, $PM_{10}$ and $PM_{2.5}$. Would be easier to read if the order is the same in both places.

Thanks for pointing it out. We've changed the order of the text to go along with the order of the figure, i.e., $NO_2$, $PM_{2.5}$, $PM_{10}$.

Line 315: all five methods are described in suppl,. but in the figure, there are results for 6 methods? Not easy to follow the names of the 6 methods in the text in the manuscript as they are not consistently defined (nor in the supplementary material).

There are 6 methods mentioned in the Supplement as well, 5 described in detail and 1 cited from another paper. To avoid confusion, we've changed the text into: "*All methods are described in Sect. S1.*" Besides, we've added corresponding method names in the figure into the text when introduced the first time.

Line 319: "Since the Romberg scheme is specifically designed to reflect measurements, providing the correct $NO_2/NO_x$ ratio, this means that the chemistry schemes are overestimating the $NO_2$ contribution when applied to annual mean concentrations." It is somewhat difficult to understand what is meant here?

There is indeed a logic jump which is unclear. We've rephrased it like this: "Since the Romberg scheme is specifically designed to reflect measurements, providing the correct $NO_2/NO_x$ ratio, *it can be regarded as the closest to the measurements. The bias differences between chemistry schemes and the Romberg scheme indicate that chemistry schemes have higher concentrations of $NO_2$ than the Romberg scheme, thus* overestimate the $NO_2$ contribution when applied to annual mean concentrations."

Table 2: within an region – should be: *within* a region

We've corrected "within an region" into "within a region".

Line 374: verses – should be *versus*

We've corrected "verses" into "versus".

Figure S5: verses – *versus* + include should be – *included*

We've corrected accordingly. We've also corrected the same typo in Figure S1.

**(EC1)**

This paper presents work around downscaling of the EMEP model. Such tools are extremely valuable for personal exposure estimates. The authors present a concise narrative of the background of the work conducted and discuss wider needs. I recommend the paper is published after the following discussion points are acted on.

Section 1, page 2 line 31 'where access to good quality emission data is available'. Can you please add more details and/or a reference? This is obviously important to understand. How is the quality and, presumably, density better in Norway? I can see this is covered in section 5.4 so please summarise and reference that section. You comment on accessibility in the proceeding sentence so is this an issue of data access?

We've added references to explain how the quality is: "where access to good quality emission related data is available. *The quality of the Norwegian emissions is summarized in Sect. 5.4 and details can be found in Sect. S4.2 of Denby et al. (2020)*".

Obtaining high resolution, 250 m, emission data from individual countries is a process in itself and requires significant time and resources. This is why we have used openly available emission proxy data that cover all of Europe and redistributed existing EMEP gridded emissions. This also allows for consistency between EMEP and uEMEP calculations.

Lines 35-40. I wonder how you could add an estimation of relative error on the downscaling predictions. Have you looked at mapping error as a function of land-use? I ask because another way to arrive at annual means is through land use regression. Can you clarify the differences?

We are unsure about what the reviewer actually wanted to see, but estimation of relative error on the downscaling predictions is validated against observations in Sect. 4. We don't know what "mapping error as a function of land-use" is. As for the land use regression (LUR) models, We haven't tested it, but we are aware of relevant studies, e.g. a European-wide LUR model study from Vizcainoand and Lavalle (2018). Though LUR is

commonly used to project the spatial concentration of atmospheric pollutants, it's a statistical method based on regression analysis. The downscaling model used in this manuscript is an atmospheric Gaussian dispersion model, which is based on physics and chemistry formulations.

Section 2 Methodology.

- Why focus on annual means and not seasonal differences? Presumably across most of Europe this changes significantly?

High-resolution simulation of uEMEP at such a large area is quite computing heavy, though parallelled in tiles. The manuscript is intended as a method description primarily, so we demonstrate the method focusing on annual means. More scientific analysis, e.g. seasonal differences across Europe, is beyond the scope of this current manuscript.

- Please can you clarify 'EMEP local fractions' – I think some of this information is provided in section 2.2 so you may wish to consider re-ordering some of the information.

Following your suggestion in the next question, we've added a new schematic figure in Sect. 2 to illustrate all steps (Fig. 1 in the revised manuscript). At the end of Sect. 2.1, we've added a new paragraph to explain 'EMEP local fractions': "*The EMEP model calculates and outputs the 'local fraction', used by the uEMEP downscaling to remove double counting of emissions (Denby et al., 2020; Wind et al., 2020). The local fraction is the contribution of emissions in one EMEP grid to itself and to its neighbouring grids. For this application only a 3 x 3 grid contribution region is calculated, though for other applications this can be much larger. By tagging the grid emissions in this way the local contribution from EMEP can be removed and replaced by the high-resolution uEMEP sub-grid calculation.*" We've also added explanations accordingly in several places of Sect. 2.2.

- may I suggest a schematic illustrating all of the steps might help the reader?

Thanks for the suggestion. We've added a new schematic figure in Sect. 2 to illustrate all of the steps.

[Figure]

Figure 1. Schematic illustration of the EMEP/uEMEP coupled modelling system.

Line 49, page 2 – '2 x 2 EMEP grids' to clarify, is this independent of the resolution of the EMEP simulations?

It's independent of the resolution of the EMEP simulations.

Table 1, page 3 – can you please clarify, are these parameters for all primary pollutants? If so, why? Please also clarify this with respect to the weight factors displayed in figure 2. It might be useful to reference section 5 at this point.

These parameters in Table 1, as well as the road weightings in Fig. 2 are for all primary pollutants. This is because the initial dispersion, emission height, and road weighting parameters are very much source specific, not species specific. We've added in the caption of Table 1: "...*for all primary pollutants.*" and caption of Fig. 2: "*Applied to all primary pollutants.*"

Section 5.1, line 216 'when calculating at station positions..'- calculating concentrations?

Yes thanks, we've rephrased it into "When calculating *concentrations* at station positions...".

Figure 10 please use black font for all axes, I found the text a little hard to read.

We've used black font for all axes and increased the font size in Fig. 10. It looks clearer now.

Page 21, line 340 – 'the largest contributor to PM is residential heating'. Please clarify whether you are referring to model outcomes or compiled data across Europe. I would like to see more references to support wider known trends in measured PM properties.

The reviewer is correct to ask for a clarification. In fact we mean 'the largest contributor to PM in the downscaled sources is residential heating'. It was not intended as a general overall statement for all concentrations of PM. We haven't analysed all sources and all precursor species to be able to make a broad statement like this. As for trends in measured PM properties, we've added a reference "Screening for High Emission Reduction Potentials for Air quality (SHERPA)" $PM_{2.5}$ urban atlas, as this provides an assessment of urban background source contributions for 150 European cities. Here they find an average contribution of 13% from primary PM from residential combustion (p. 13).

We've corrected this in the text around line 340: "The largest contributor to PM *in the downscaled sources* is residential heating*, with contributions 19% and 16% for $PM_{2.5}$ and*

*PM$_{10}$, respectively (Table 2). This is inline with other estimates of residential combustion in Europe. Thunis et al. (2017) calculated a contribution of 13% from residential combustion from primary PM$_{2.5}$ averaged over 150 European cities, without downscaling. Population is used as a downscaling proxy for the residential source, but...".*

Code and data availability – please can you also add links to the EMEP and IFS version used, with relevant configuration files.

We've added the link to the EMEP model version and the configuration file used in this study. We run the IFS model version Cycle 40r1 (ECMWF-IFS cy40r1) to generate the meteorological files, as input for the EMEP model. As far as we know, the IFS model is not open-source, so that users need to request a licence from ECMWF. Therefore we can't provide a link to the IFS model used with relevant configuration file. However, all input files can be shared upon request, including the meteorological input files that are generated by the IFS model.

In "Code and data availability", we've added the following text: "*The EMEP model version rv4.36 used in this study can be found at https://github.com/metno/emep-ctm. The configuration file of the EMEP model is archived at https://doi.org/10.5281/zenodo.5648144. Other model input files can be shared upon request, including the meteorological input files that are generated by the IFS model version Cycle 40r1 (ECMWF-IFS cy40r1).*" The EMEP model version has also been corrected from rv4.35 to rv4.36 in Sect. 2.1.

Title – please change uEMEP_v6 to uEMEP v6. I would also consider adding the year, since there will be variation during periods of large scale change?

The uEMEP description paper (Denby et al., 2020) named the model as "uEMEP_v5", so "uEMEP_v6" just keeps the consistency. When we submitted that article to GMD the editor requested this model naming. We can change to "uEMEP v6", if very necessary, but we think this is a question for the editor and the journal's requirements. So far, the uEMEP model only

uses numbers to denote versions after large scale changes. The exact code version used for these calculations has been given a doi, as requested by the journal.

**(RC2)**

This publication presents a downscaled version of the EMEP model and its application to Europe for air quality pollutants. The topic is very interesting and certainly relevant with respect to the scope of the journal. The methodological approach is sound and the work is well structured and well presented. I list below a few comments.

Comments:

1. The introduction dives directly into the proposed uEMEP downscaling. I would suggest starting with a paragraph to provide some context and explain why we actually need such a downscaling.

   We've now added why we need such downscaling following on from L28: "...so that we can have air quality modelling at street-level all over Europe. *Modelling at high resolution provides a better assessment of air quality mitigation strategies in Europe, as well as improved population exposure estimates for use in health impact studies.*"

   While the Authors state (L107) that "uEMEP downscales only primary pollutants", downscaled results for O3 are discussed and shown. A few lines to explain how secondary pollutants like O3 can be downscaled would clarify a possible confusion.

   How this is done is explained in the rest of section 2.4 but we have added a little bit more to the sentence on L107 that may help: "uEMEP downscales only primary pollutants. It is thus necessary to apply chemistry parameterisations to the $NO_x$ and *EMEP* $O_3$ concentrations to derive *downscaled* $NO_2$ *and* $O_3$ *concentrations*."

2. Orography variations can be important over two 0.1deg grid cells. Can the authors comment on the applicability of their Gaussian approximation to hilly cities?

   Gaussian approximations have of course their limitations and the quality of the results will reflect, amongst other things, the complexity of the terrain. In essence we use winds from 0.1° meteorological fields so the accuracy of these in complex terrain can always be questioned. We have not assessed this, nor have we assessed the impact of buildings that are also not included in the modelling. We cannot provide a quantitative answer to the reviewers question.

3. A few remarks regarding Table 2:

1. Caption: an region à a region

   Corrected.

2. Please add that the region of +/- 0.1 deg also represents 2x2 grid cells.

   We've modified it into "... within a region of 2 x 2 EMEP grids (± 0.1°) …"

3. If my understanding is correct, the source column should precise that only primary contributions are considered (e.g. primary traffic…) and that non-local EMEP includes all secondary contributions as well.

   We've made it clear in the text that "uEMEP local contributions are from *primary* emissions…", also "Non-local EMEP model contributions are all emissions from outside this region, for the downscaled sources, as well as all other primary and precursor emission sources from within this region that are not downscaled."

   It is unclear how the non-local EMEP is obtained from the 2x2 EMEP local fractions. If summed up, how is double-counting avoided? Indeed, the non-local fraction of one given cell reaching the neighbouring cell will be counted as non-local while it is actually local within the 2x2 area, right?

   At every calculation point in uEMEP the local contribution from the surrounding 2 x 2 region in EMEP is removed to provide the non-local contribution. The EMEP local fraction calculation tracks over several EMEP grids, so it is not just the grid itself that the local contribution is removed from, but also neighbouring grids as required. We've added a schematic figure in Sect. 2 to illustrate it.

4. Can the Authors detail the 2x2 grid cell kernel calculation (L48-49)? For an emission located within a given EMEP cell, say near its SW corner, how are these 2x2 cells defined?

   We've added a new schematic figure to illustrate all steps in Sect. 2. In Sect 2.2, we have changed text to: "*For these simulations this region corresponds to 2 x 2 EMEP*

*grids, i.e. within an area that is ± 0.1° in both latitude and longitude. This ensures that no matter where the uEMEP calculation sub-grid is placed that the moving window region will always be covered by the 3 x 3 local fraction region*".

Technical remarks

1. L45: even though detailed in other references (listed), some explanations on how the EMEP local fraction is calculated would be helpful in this work.

   We've added a new schematic figure in Sect. 2 to illustrate all steps (Fig. 1 in the revised manuscript). At the end of Sect. 2.1, we've added a new paragraph to explain 'EMEP local fractions': "*The EMEP model calculates and outputs the 'local fraction', used by the uEMEP downscaling to remove double counting of emissions (Denby et al., 2020; Wind et al., 2020). The local fraction is the contribution of emissions in one EMEP grid to itself and to its neighbouring grids. For this application only a 3 x 3 grid contribution region is calculated, though for other applications this can be much larger. By tagging the grid emissions in this way the local contribution from EMEP can be removed and replaced by the high-resolution uEMEP sub-grid calculation.*" We've also added explanations accordingly in several places of Sect. 2.2.

2. L56: Can the authors comment on the height of the first vertical layer (50m). Does this impact the downscaled results?

   The height of the lowest layer in EMEP has some, but little, impact on the results since the lowest layer concentrations calculated by EMEP are removed and replaced by uEMEP in the 2 x 2 grid region surrounding each sub-grid. There are many details such as this that we could have provided sensitivity studies for, but we have limited these to the studies provided.

3. L85: Is the split in tiles motivated by CPU gains only or does it improve the efficiency for other aspects?

We can run multiple simulations in parallel, one simulation for each tile. So it's mostly motivated by making full use of the computing power.

4. L93: "Weights are based on Norwegian average road situations". Can the authors comment on the validity of this assumption when applying it to other countries

At the end of the first paragraph of Sect 2.3, we commented that: "Sensitivity tests with alternative weighting, see Sect. 5.2, show the choice of weighting does impact on the results but that the current choice provides close to optimal spatial correlation when compared to measurements."

5. L99: The choice of population as proxy to redistribute residential heating emissions is known to lead to important issues for some cities (as shown later in the document). Is this initial choice related to data availability?

It should be noted that we redistribute each grid's emission by population within that grid. This does not give the same problems as distributing a country's emissions using population. If some cities are known to have reduced residential combustion in the EMEP emissions inventory then they will be reduced there already. As shown in Sect. 5.3, we tested a range of other proxies and variants of the population density as proxy. Population data is indeed very available data for this proxy and was the first choice because 1) it is available everywhere and 2) it does spatially delineate where the emissions come from, even though the emissions may not be directly related to the actual population. That is why we have also tried building density and building density masked with population and a number of population power factors that reduce or enhance the impact of population density for scaling emissions. The results in Section 5.3 address this but do not provide a convincing 'best proxy'.

6. L171 and L334: The spatial representativeness of some traffic stations, especially those located in street canyons is lower than 25 meters. Can the authors comment on their choice to keep those stations anyway for the comparison?

We don't exactly know how many traffic sites are street canyon sites. Reporting to the EU Commission (e-reporting) on siting of traffic stations is generally quite incomplete. Metadata provided in e-reporting was analysed for the EU commission in 2018 (Tarrasón et al., 2021). It indicated that the majority of stations are not street

canyon sites, though the exact number was unclear and most countries do not report this at all.

7. L169 and others: Note that the correlation is actually "r" not "r2". The latter is referred to as the coefficient of determination. Please adapt the text or figures accordingly.

In the first paragraph of Sect. 4, we've corrected the expression as "Results focus on the spatial correlation, *expressed in terms of the coefficient of determination* ($r^2$)*, and* …". We've also made sure that it is "spatial correlation" instead of "correlation" in the text, and put in "coefficient of determination" in the related figure captions.

8. Figures 10 and 12: please explain which station or station averages are shown.

We've added an explanation in the captions of Fig. 10 and Fig. 12: "The results are based on the European calculations and all available Airbase stations are included."

9. L229: "the" is doubled twice!

Corrected.

10. L311: Can the Authors comment on the robustness of this fitting for other years?

The fitting parameters have only been assessed for European stations for the year 2017, though several years have been fitted for Norwegian stations. These Norwegian fits show a robust, though slightly different to the Europe, relationship since $O_3$ levels are built into the fitting parameters. The fitting parameters are expected to be slightly different in different years and for different datasets due to changes in $NO_2/NO_x$ emission ratios for traffic and also possibly reduced $NO_X$ emissions and varying ozone levels. If such fitted data were to be used for later years, e.g. 2021, then these fits would need to be updated. We do not intend to use the Romberg scheme for further work but find it instructive for this study.

**References**

Denby, B. R., Gauss, M., Wind, P., Mu, Q., Grøtting Wærsted, E., Fagerli, H., Valdebenito, A., and Klein, H.: Description of the uEMEP_v5 downscaling approach for the EMEP MSC-W chemistry transport model, Geosci. Model Dev., 13, 6303–6323, https://doi.org/10.5194/gmd-13-6303-2020, 2020.

Tarrasón, L., Hak, C., Soares, J., Røen, H., Ødegård, R., Green, J., and Marsteen, L.: Assessing the spatial representativeness of air quality sampling points: Application of siting criteria and sampling point classification – Task 3 report, Ricardo Group, Ref: ED 11492 – Final Task 3, 2021.

Lefebvre, W., Van Poppel, M., Maiheu, B., Janssen, S., Dons, E.: Evaluation of the RIO-IFDM-street canyon model chain, Atmos. Environ., 77, 325–337, https://doi.org/10.1016/j.atmosenv.2013.05.026, 2013.

Vizcaino, M. and Lavalle, C.: Development of European $NO_2$ Land Use Regression Model for present and future exposure assessment: Implications for policy analysis, Environ. Pollut., 240, 140-154, https://doi.org/10.1016/j.envpol.2018.03.075, 2018.

Thunis, P., Degraeuwe, B., Peduzzi, E., Pisoni, E., Trombetti, M., Vignati, E., Wilson, J., Belis, C. and Pernigotti, D.: Urban $PM_{2.5}$ Atlas: Air Quality in European cities, EUR 28804 EN, Publications Office of the European Union, Luxembourg, 2017.